# Human-like Supramodal Concept Learning Boosts Emotion Recognition

## Abstract

Multimodal emotion recognition has shown promise but is often hindered by the complexity of integrating heterogeneous sensory inputs. Intriguingly, the human brain addresses this challenge through abstract, modality-independent emotion schemas, known as supramodal emotion concepts, which are learned gradually from emotional experiences across different sensory modalities. Here, we propose a learning strategy to construct supramodal emotion concepts across vision, text, and audio. Each modality's data repeatedly passes through a shared emotion encoder and its corresponding modality-specific non-emotion encoder in a decoupling framework, extracting modality-independent emotion representations. Inspired by hippocampal replay in humans, these representations are aggregated from a memory pool during downstream emotion recognition to form supramodal emotion concepts. We demonstrate the effectiveness of this approach in multiple settings: (1) a lightweight image-based model achieves state-of-the-art results on several benchmark datasets with lower complexity than existing unimodal methods; (2) unimodal models using vision, text, or audio from video clips achieve performance comparable to multimodal models; and (3) concept-guided multimodal models further improve performance, surpassing current state-of-the-art.

## 1 Introduction

Emotion recognition (ER) is essential for artificially intelligent machines to understand human affective states. Recently, ER has flourished in the deep learning community and achieved impressive progress, showing great potential for widespread applications, including disease detection (Yeung, 2022) and intelligent tutoring systems (Petrovica et al., 2017). However, existing CNN- (Li et al., 2017a; Ruan et al., 2021) and Transformer-based (Xue et al., 2021; Li et al., 2021b) models for unimodal ER still fall short of human-level performance, possibly due to their limited ability to capture the richness and complexity of emotional signals in real-world contexts (Goel et al., 2024).

Multimodal fusion has become a major focus in ER research, as different modalities within the same context are often complementary to each other, providing additional cues that may facilitate robust emotional understanding (Zadeh et al., 2017; Li et al., 2023; Wang et al., 2023; 2024). Although mainstream multimodal fusion methods have demonstrated promising developments for ER, they face two major challenges: 1) the need to address the heterogeneity between different modalities, which greatly increases model complexity (Li et al., 2023; Hazarika et al., 2020), and 2) the requirement for paired image-text-audio data to provide complementary emotional information, which complicates the acquisition of high-quality training data (Wang et al., 2024; 2023). In contrast, humans demonstrate strong and consistent performance in emotion classification tasks regardless of input modality—image, text, or audio. This capability may arise from the engagement of the ventromedial prefrontal cortex (vmPFC), a brain region implicated in encoding supramodal emotion concepts (Lettieri et al., 2024; Camacho et al., 2023).

Recent neurobehavioral experiments in the field of affective neuroscience suggest that supramodal emotion concepts may emerge through mechanisms such as hippocampal replay, which repeatedly reactivates and reorganizes emotional representations learned from diverse multimodal sensory experiences (Carr et al., 2011; Rothschild et al., 2017). Through this iterative process, supramodal emotion concepts are formed and subsequently stored in the brain's hierarchical memory system, particularly involving the vmPFC (Rolls, 2023). According to the theory of constructed emotion,

these concepts serve as priors for predicting emotional instances (Barrett, 2017). This indicates that emotion perception in the brain's sensory system relies not only on sensory features but also on the supramodal emotion concepts (Brooks et al., 2019). As a result, this specialized sensory coding facilitates rapid responses to emotional cues in the human brain, regardless of whether the input is image, audio, or text. Furthermore, it offers an opportunity to design a brain-inspired approach to supramodal concept learning, which may enhance emotion recognition by emulating the human ability to generalize emotional understanding across modalities.

In this work, we draw inspiration from the hippocampal replay mechanism to extract abstract concepts from multimodal signals. We first design a shared emotion encoder that learns modality-independent emotional representations from images, text, and audio in a sequential and iterative manner, mirroring how humans gradually acquire emotional knowledge through repeated multimodal experiences. One strength of this iterative training scheme is that it does not require paired image–text–audio data from the same video clips. Building on these representations, we introduce a replay-inspired strategy to construct supramodal emotion concepts, which are then used to regularize downstream emotion recognition models. The effectiveness of our framework is demonstrated in the experimental section.

The contributions of this work can be summarized as:

- We propose a sequential and iterative learning strategy that mimics how humans gradually acquire emotional representations from diverse modalities, enabling the extraction of modality-independent features.

- We introduce a brain-inspired replay strategy to construct supramodal emotion concepts at a high level of abstraction, providing effective guidance for downstream emotion recognition models.

- We conduct extensive experiments on both benchmark unimodal datasets and curated multimodal datasets, demonstrating that concept-guided unimodal and multimodal models achieve strong performance, validating the effectiveness and generalizability of supramodal emotion concepts.

## 2 RELATED WORKS

### 2.1 MULTIMODAL EMOTION RECOGNITION

Multimodal emotion recognition aims to integrate complementary emotional information from the same video clip to achieve higher emotion recognition performance than unimodal approaches. Currently, multimodal fusion networks for emotion recognition can be divided into two categories: the complete (Zadeh et al., 2017; Li et al., 2023; Tsai et al., 2019a) and incomplete multimodal learning (Wang et al., 2023; 2024). The former typically needs to address the challenges of information redundancy and heterogeneity across different modalities before fusion (Li et al., 2023; Hazarika et al., 2020). Therefore, some works employ feature decoupling to facilitate more effective fusion across multimodal representations. This approach involves creating two pathways: one for processing modality-independent components and the other for modality-dependent components, which are then combined for emotion recognition. Incomplete multimodal learning is more flexible in terms of data requirements, allowing for missing modality. The mainstream method includes recovering missing modalities through generative models with the assistance of available modalities (Wang et al., 2023; 2024). The generated modalities are subsequently integrated with the existing modalities for the task of emotion recognition. Unlike previous studies that rely on multimodal fusion frameworks, our approach is flexible in its modality requirements. We employ a sequential and iterative learning strategy to extract modality-independent emotional representations from diverse multimodal inputs.

### 2.2 CONCEPT LEARNING

One active research topic in brain-inspired intelligence is how to extract abstract concept representations from deep neural networks (DNNs). For example, CDP (Zeng et al., 2019) and CRPN (Lu et al., 2024) extract conceptual information from texts or images, projecting feature representations

into a conceptual space. While these concepts derived from DNNs can improve model performance, their unimodal design may still result in modality-dependent attributes in the extracted conceptual information. To address these limitations, recent approaches have shifted towards multimodal frameworks. Both MoMo (Chada et al., 2023) and OneLLM (Han et al., 2024) employ a unified framework to extract representations from different modalities while being data, memory and runtime efficient. However, they require additional constraints on the model, such as the cross-modality gradient accumulation to prevent catastrophic forgetting. Although our approach shares similarities with CRPN (Lu et al., 2024) in using orthogonality with non-emotion encoders to disentangle emotion concepts, we further introduce a sequential learning strategy and a replay-inspired mechanism to extract high-quality supramodal concepts from diverse modalities. Unlike CRPN, which is limited to image data, our framework enhances both the abstraction level and generalization capability of concepts.

## 3 METHOD

The proposed human-like framework consists of two phases: supramodal concept learning and supramodal concept evaluation. In the learning phase, a shared emotion encoder and three modality-specific non-emotion encoders are employed to isolate abstract, modality-independent emotion representations from visual, text, and auditory inputs, thereby enabling the construction of supramodal emotion concepts in the human brain's vmPFC. In the evaluation phase, we use the learned supramodal concepts to regularize downstream emotion recognition models, assessing whether these concepts effectively enhance model performance.

### 3.1 SUPRAMODAL CONCEPT LEARNING

We consider three modalities, *i.e.*, image (I), text (T) and audio (A) in the concept learning phase. Fig. 1 depicts the learning framework designed to extract modality-independent emotion representations, which form the basis for constructing supramodal emotion concepts. This framework includes the CLIP image and text encoders, the CLAP audio encoder, a shared emotion encoder, three modality-specific non-emotion encoders and an emotion classifier. To effectively extract emotion representations, we establish a two-stage training pipeline comprising multimodal joint learning and sequential cross-modal learning. Detailed descriptions are presented in the following sections.

**Projecting different modal inputs into an embedding space.** From a deep learning perspective, the pre-trained CLIP (Radford et al., 2021) and CLAP (Elizalde et al., 2023) models demonstrate robust representational capabilities due to extensive training on large-scale datasets. From a computational neuroscience perspective, Transformer-based models outperform CNNs in capturing neural response patterns in mid-to-high-level brain regions, including the vmPFC (Caucheteux et al., 2023), which stores the supramodal emotion concepts. Therefore, we employ the Transformer-based CLIP image encoder, CLIP text encoder, and CLAP audio encoder to project images, text, and audio into a 512-dimensional embedding space to obtain modal features $\boldsymbol{f}_I$, $\boldsymbol{f}_T$ and $\boldsymbol{f}_A$, respectively. To mitigate the computational cost of full fine-tuning, we adopt a LoRA fine-tuning approach (Hu et al., 2021), freezing pre-trained weights and injecting trainable rank decomposition matrices into each Transformer layer.

**Extracting modality-independent emotion features from different modal features.** Inspired by neuroscience studies (Haxby et al., 2000; Zhang et al., 2023) on two distinct neuroanatomical pathways in the human brain that process variable features (e.g., emotions) and invariant features (e.g., identity, age, and gender), we developed various types of encoders: one emotion encoder and three modality-specific non-emotion encoders to simultaneously process the modal features. Specifically, different modal features are processed through a shared emotion encoder $E^{emo}$ to extract modality-independent emotion features. Additionally, we employ a modality-specific non-emotion encoder $E_m^{non}$, where $m \in \{I, T, A\}$, to capture modality-specific, non-emotional variations within each modality. Formally,

$$\boldsymbol{f}_m^{emo} = E^{emo}(\boldsymbol{f}_m), \boldsymbol{f}_m^{non} = E_m^{non}(\boldsymbol{f}_m). \tag{1}$$

We first apply soft orthogonality to minimize information redundancy between emotion and non-emotion features. Second, an emotion classifier is added after the shared encoder to focus learning on emotion-relevant information. Let $\boldsymbol{F}_m^{emo} = [\boldsymbol{f}_{m_1}^{emo}, \boldsymbol{f}_{m_2}^{emo}, ..., \boldsymbol{f}_{m_N}^{emo}]^T$ and $\boldsymbol{F}_m^{non} =$

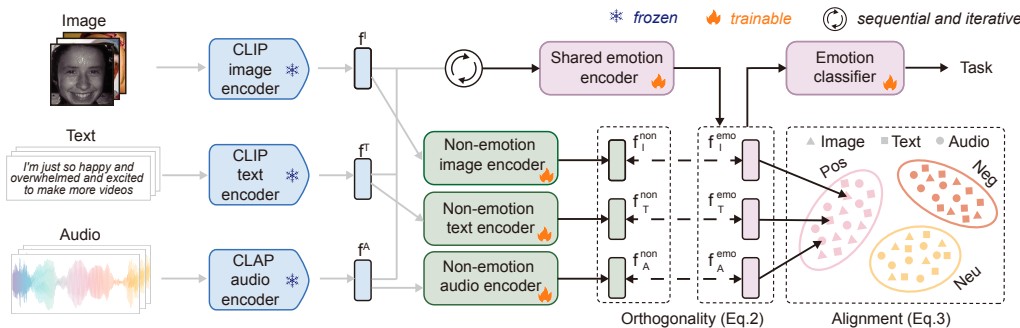

Figure 1: The framework for learning emotion representations. In the multimodal joint learning stage, supervised contrastive learning is employed to ensure that features representing the same emotion across different modalities are closely aligned. In the sequential cross-modal learning stage, data from each modality are iteratively fed into the model, mimicking the way humans are repeatedly exposed to inputs from different modalities.

$[\boldsymbol{f}_{m_1}^{non}, \boldsymbol{f}_{m_2}^{non}, ..., \boldsymbol{f}_{m_N}^{non}]^T$, respectively, where $N$ is the batch size, $m_i$ denotes the modality of the $i$-th sample. We define an orthogonal loss $\mathcal{L}_{orth_m}$ as

$$\mathcal{L}_{orth_m} = \left\| \boldsymbol{F}_m^{emoT} \boldsymbol{F}_m^{non} \right\|_F^2, \tag{2}$$

where $\|\cdot\|_F^2$ is the squared Frobenius norm.

**Multimodal joint learning.** While CLIP effectively aligns image and text representations, audio features $\boldsymbol{f}_A$ obtained from the CLAP audio encoder, although having the same dimensionality as $\boldsymbol{f}_I$ and $\boldsymbol{f}_T$, may not reside in the same semantic space. This misalignment can hinder the extraction of modality-independent emotional information using the shared emotion encoder. To address this, we apply the supervised contrastive learning (Khosla et al., 2021) to emotion features across modalities. Specifically, emotion features corresponding to the same emotion across modalities are pulled closer than those representing different emotions within the same modality, with a margin of $\alpha$. Therefore, we define a contrastive loss as

$$\mathcal{L}_{contrast} = \frac{1}{|S|} \sum_{(i,j,k)\in S} max(0, \alpha - \cos(\boldsymbol{f}_{m_i}^{emo}, \boldsymbol{f}_{m_j}^{emo}) + \cos(\boldsymbol{f}_{m_i}^{emo}, \boldsymbol{f}_{m_k}^{emo})), \tag{3}$$

where we collect a triple tuple set $S = \{(i,j,k) | m_i \neq m_j, m_i = m_k, c_i = c_j, c_i \neq c_k\}$. Here, $c_i$ is the emotion label of the $i$-th sample, $cos(\cdot, \cdot)$ represents the cosine similarity between two emotion features.

In summary, during multimodal joint learning, the total loss function is given below

$$\mathcal{L}_{joint} = \mathcal{L}_{task} + \beta_1 \mathcal{L}_{contrast} + \beta_2 \mathcal{L}_{orth}, \tag{4}$$

where $\mathcal{L}_{task}$ is the sum of emotion task-related loss (*i.e.,* multi-class cross-entropy loss) for each modality, $\mathcal{L}_{orth} = \sum_{m\in\{I,T,A\}} \mathcal{L}_{orth_m}$, the coefficients $\beta_1$ and $\beta_2$ are balanced factors.

**Sequential cross-modal learning.** At this stage, we adopt a sequential training strategy in which each modality is exposed iteratively to the model, enabling the shared emotion encoder to learn modality-independent emotion representations without requiring paired image-text-audio data. For example, when the current inputs are images, they are processed through the CLIP image encoder, the shared emotion encoder, and finally the classifier for emotion recognition. Simultaneously, the images are also processed by the non-emotion image encoder to filter out any emotion-irrelevant information. The effect of training cycle duration is further examined in the ablation studies.

In summary, during sequential cross-modal learning, the whole loss function for each training cycle, involving a single modality, is defined as

$$\mathcal{L}_{sequential_m} = \mathcal{L}_{task_m} + \gamma \mathcal{L}_{orth_m}, \tag{5}$$

where $\gamma$ is the balanced factor.

**Replay-inspired construction of supramodal emotion concepts.** After learning the modality-independent emotion representations, we freeze the model and introduce a replay-inspired mechanism to construct supramodal emotion concepts. In the hippocampus, replay is not a uniform reactivation of all past experiences but shows selectivity, giving preference to salient events (Huelin Gorriz et al., 2023). To mimic this, we build a high-confidence feature pool for each modality by retaining the top 20% correctly predicted samples per class. Replay also supports generalization, integrating across diverse experiences rather than mechanically repeating single episodes (Whittington et al., 2018). To capture this property, when processing an input $\boldsymbol{x}_i$ with an emotion label $y$ in a downstream task, we randomly sample $k$ high-confidence features from each modality's pool corresponding to $y$ and aggregate them to form a supramodal emotion concept:

$$\boldsymbol{f}_i^{ec} = \frac{1}{|M|} \sum_{m \in M} \frac{1}{k} \sum_{j=1}^{K} \boldsymbol{f}_{m_j}^{emo}, \tag{6}$$

Here, $M$ represents the set of available modalities (which can include $I$, $T$, $A$). $\boldsymbol{f}_{m_j}^{emo}$ denotes the emotion features of the $j$-th sample from modality $m$. We set $k = 8$ in this paper.

### 3.2 SUPRAMODAL CONCEPT EVALUATION

In the concept evaluation phase, we assess the effectiveness of supramodal emotion concepts through downstream emotion recognition. For unimodal evaluation, a single modality is processed by its encoder to obtain a concept-guided feature (i.e., $\boldsymbol{f}_i^{uni}$). For multimodal evaluation, features from all modalities are first fused and then passed through a fully connected layer to obtain a concept-guided feature (i.e., $\boldsymbol{f}_i^{multi}$). In both cases, the corresponding supramodal concept $\boldsymbol{f}_i^{ec}$ is simultaneously derived and used to guide the downstream models.

We design a similarity loss $\mathcal{L}_S$ to encourage the downstream models to extract rich emotional information from the supramodal emotion concepts,

$$\mathcal{L}_S = \frac{1}{N} \sum_{i=1}^{N} (\boldsymbol{f}_i^{ec} - \boldsymbol{f}_i^{down})^2. \tag{7}$$

where $\boldsymbol{f}_i^{down}$ denotes either $\boldsymbol{f}_i^{uni}$ or $\boldsymbol{f}_i^{multi}$.

In summary, during the training of the downstream model, the total loss function is defined as

$$\mathcal{L}_{down} = \mathcal{L}_{CE} + \lambda \mathcal{L}_S, \tag{8}$$

where $\mathcal{L}_{CE}$ is the multi-class cross-entropy loss, $\lambda$ is the balanced factor.

## 4 EXPERIMENT

### 4.1 IMPLEMENTATION DETAILS

**Datasets.** To mitigate learning biased representations due to imbalances in data scales across different modalities, we curate a multimodal dataset for supramodal concept learning. The image dataset is RAF-DB(Li et al., 2017b), an in-the-wild facial emotion dataset. The text dataset combines samples from CMU-MOSI(Zadeh et al., 2016) and CMU-MOSEI(Zadeh et al., 2018b). The audio dataset comprises three high-quality emotion-annotated datasets: IEMOCAP(Busso et al., 2008), MELD(Poria et al., 2018), and RAVDESS(Livingstone & Russo, 2018). For the supramodal concept evaluation, we use three in-the-wild facial emotion datasets: RAF-DB, AffectNet(Mollahosseini et al., 2017) and FED-RO(Li et al., 2018) and two emotion datasets based on video clips: CMU-MOSI(Zadeh et al., 2016) and CMU-MOSEI(Zadeh et al., 2018b). In summary, each sample in the image and audio datasets is labeled as one of seven basic emotions (happiness, anger, sadness, fear, disgust, surprise and neutral), while each sample in the text dataset is labeled as either positive, neutral, or negative. We evaluate 7-class accuracy for the image and audio datasets unless otherwise specified, and 3-class accuracy for the text datasets. Further details are provided in Appendix A.

**Training Details.** During supramodal concept learning, we employ pre-trained CLIP ViT-B/32 and CLAP HTSAT models, fine-tuned with LoRA (rank 6, alpha 36, dropout 0.2). The shared emotion

encoder has two fully connected (FC) layers: the first projects 512-dimensional input features from CLIP or CLAP to 256 dimensions with batch normalization and ReLU, and the second maintains 256 dimensions with the same normalization and activation. Each modality also has a non-emotion encoder with the same architecture. The emotion classifier is a single FC layer mapping the 256-dimensional emotion features to the number of emotion categories.

During supramodal concept evaluation, we consider three settings. (1) An image-based model that uses ResNet-18 (He et al., 2016) as the backbone. (2) Unimodal emotion recognition from vision, text, or audio extracted from video clips, implemented with the Transformer architecture from (Li et al., 2023). (3) Multimodal emotion recognition, where 256-dimensional features from all three modalities—each obtained using the same modality-specific Transformers as in (Li et al., 2023)—are concatenated into a 768-dimensional representation. These resulting features are further processed by an additional FC layer with 256 neurons to obtain the representations used for concept regularization. Details of modality preprocessing are in Appendix B.

During multimodal joint learning, to align emotion features across modalities, we harmonize their labels. Following (Li et al., 2021a), fear, disgust, sadness, and anger in the image and audio datasets are grouped as negative emotions, happy as positive, neutral unchanged, and surprise excluded. During sequential cross-modal learning,original emotion labels are used. To evaluate potential loss of discrimination among negative emotions when collapsing from 7 to 3 classes for text alignment, we compared image and audio performance under both 7- and 3-class settings (see Appendix C).

All experiments are conducted using PyTorch on two NVIDIA GeForce RTX 4090 GPUs with a batch size of 64. We use AdamW (weight decay 1e-5, initial learning rate 1e-4) with a cosine scheduler of period 5. During supramodal concept learning, multimodal joint learning is trained for 10 epochs and sequential cross-modal learning for 100 epochs until convergence. Concept-guided unimodal models are trained for 40 epochs. The hyperparameters are set as $\alpha = 0.2$, $\beta_1 = 0.05$, $\beta_2 = 0.1$, $\gamma = 0.1$, and $\lambda = 0.5$, achieving the best performance in this work.

### 4.2 COMPARISON WITH THE STATE-OF-THE-ART

We compare the performance of the image-based emotion recognition model, guided by supramodal emotion concepts, with the current state-of-the-art methods for facial emotion recognition. As listed in Table 1, our model is among the best on both the RAF-DB and the AffectNet data sets. When we combine the RAF-DB and the AffectNet data sets (*i.e.*, R & A) for training, our model achieves the best performance when tested using the independent FED-RO data set and is also among the lightest (*i.e.*, 11M parameters). The pretraining datasets used for evaluating each method on FED-RO are listed in Appendix D.

We train concept-guided unimodal models using a single modality (vision, text, or audio) from video clips and compare their performance with state-of-the-art multimodal emotion recognition methods. As shown in Table 2, even with only one modality, our models achieve performance comparable to multimodal methods, demonstrating the effective guidance of supramodal emotion concepts and the capability of "borrowing of strength." We further train concept-guided multimodal models and observe that these models surpass current state-of-the-art results, further highlighting the benefit of supramodal concept guidance. To confirm the robustness of these improvements, we conduct paired t-tests comparing multimodal models with and without concept guidance, revealing a statistically significant advantage for our method (see Appendix E).

### 4.3 ABLATION EXPERIMENTS

**Evaluation of the two-stage training pipeline for the supramodal concept learning.** We compare the performance of our two-stage training pipeline with models trained using only a single stage. To ensure a fair comparison, the single-stage models are trained for the same number of epochs as the full two-stage pipeline. As shown in Table 3, both single-stage models underperform, demonstrating the effectiveness of combining both stages to extract robust modality-independent emotion features.

**Evaluation of the duration of training cycle for each modality during supramodal concept learning.** We assess the impact of training cycle duration for each modality in the sequential cross-modal training, with batch- and epoch-based results shown in Table 4. First, a too-short cycle

Table 1: Comparison of our concept-guided image-based model with state-of-the-art methods on RAF-DB, AffectNet and FED-RO (%). #Params – number of parameters.

| Method | RAF-DB | AffectNet | FED-RO | #Params |
|---|---|---|---|---|
| VGG16 (Simonyan & Zisserman, 2014) | 85.16 | 58.21 | 63.49 | 138M |
| ResNet18 (He et al., 2016) | 86.08 | 59.15 | 65.32 | 11M |
| gACNN (Li et al., 2018) | 85.07 | 58.78 | 66.50 | 224M |
| SPWFA-SE (Li et al., 2020) | 86.31 | 59.23 | 67.25 | 21M |
| RAN (Wang et al., 2020b) | 86.90 | 59.50 | 67.98 | 11M |
| SCN (Wang et al., 2020a) | 87.03 | 60.23 | 68.24 | 11M |
| DMUE (She et al., 2021) | 89.42 | 63.11 | - | >25M |
| CRPN (Lu et al., 2024) | 89.71 | 63.06 | 71.00 | 11M |
| TransFER (Xue et al., 2021) | 90.91 | 66.23 | - | >25M |
| Ours | **91.02** | 65.51 | **76.00** | 11M |

Table 2: Comparison of our concept-guided models and state-of-the-art multimodal fusion methods on the CMU-MOSI and CMU-MOSEI datasets (%).

| Method | CMU-MOSI | | CMU-MOSEI | |
|---|---|---|---|---|
| | $ACC_2$ (%) | F1 (%) | $ACC_2$ (%) | F1 (%) |
| MFM (Tsai et al., 2019b) | 78.1 | 78.1 | - | - |
| Graph-MFN (Zadeh et al., 2018a) | - | - | 76.9 | 77.0 |
| MCTN (Pham et al., 2019) | 79.3 | 79.1 | 79.8 | 80.6 |
| RAVEN (Wang et al., 2019) | 78.0 | 76.6 | 79.1 | 79.5 |
| MulT (Tsai et al., 2019a) | 83.0 | 82.8 | 82.5 | 82.3 |
| PMR (Lv et al., 2021) | 83.6 | 83.4 | 83.3 | 82.6 |
| FDMER (Sun et al., 2023) | 84.6 | 84.7 | - | - |
| MICA (Liang et al., 2021) | - | - | 83.7 | 83.3 |
| DMD (Li et al., 2023) | 86.0 | 86.0 | 86.6 | 86.6 |
| Ours (Audio) | 80.5 | 80.2 | 80.6 | 80.2 |
| Ours (Text) | 85.8 | 85.8 | 86.1 | 85.7 |
| Ours (Vision) | 86.2 | 86.0 | 86.4 | 85.8 |
| Ours (Multimodal) | **87.4** | **87.2** | **87.5** | **87.5** |

harms performance across all modalities. For example, when the cycle equals one batch (1st row), the model switches rapidly between image, text, and audio, preventing reliable learning and causing representation drift. Increasing the cycle, particularly to 100 batches, improves performance. Second, a too-long cycle leads to forgetting previously learned modalities. When the cycle equals five epochs (6th row), the model, concluding with audio, shows reduced performance on image and text. Setting the cycle to one epoch effectively balances under-training and catastrophic forgetting.

**Evaluation of the impact of different concept-guided methods on the unimodal models.** Both the image-based model and the unimodal emotion recognition models based on vision, text, or audio are evaluated under various types of concept guidance, with the former trained on the R & A

Table 3: Comparison of the two-stage training pipeline and single-stage training on our curated multimodal emotion datasets.

| Modality | Multimodal joint learning | Sequential cross-modal learning | Two-stage |
|---|---|---|---|
| Image | 87.33 | 90.20 | **91.59** |
| Text | 70.25 | 73.67 | **75.42** |
| Audio | 53.81 | 56.84 | **60.97** |

Table 4: The impact of the duration of training cycle on the accuracy (%) for each modality during supramodal concept learning.

| Replay type | Image | Text | Audio |
|---|---|---|---|
| 1 batch | 89.89 | 73.01 | 55.39 |
| 10 batch | 90.14 | 73.81 | 56.15 |
| 100 batch | 91.21 | 74.89 | 58.73 |
| 1 epoch | **91.59** | **75.42** | **60.97** |
| 2 epoch | 91.03 | 74.21 | 60.75 |
| 5 epoch | 88.62 | 73.35 | 60.88 |

dataset and tested on the independent FED-RO dataset, and the latter separately trained and tested on the CMU-MOSI and CMU-MOSEI datasets. We show the experimental results in Table 5. For the FED-RO dataset, performance is evaluated using $ACC_7$, while for MOSI and MOSEI, $ACC_2$ is used as the evaluation metric. (1) Incorporating supramodal concept guidance consistently improves performance compared to models without guidance (1st vs. other rows). (2) Increasing modalities during concept construction boosts performance (2nd-8th rows). Emotion concepts derived from modalities other than the one used by the unimodal model still provide valuable guidance. (3) Removing replay-inspired selectivity, by sampling features directly from modality-independent representations instead of high-confidence pools, degrades performance (the 9th row). (4) Removing replay-inspired generalization, by replacing the averaging operation with a simple random selection of one emotional feature from high-confidence pools, also reduces accuracy (the 10th row). Together, these results demonstrate the effectiveness of replay-inspired supramodal emotion concept construction.

**Evaluation of the abstraction and generalizability of the learned concepts.** To test whether supramodal concepts capture abstract and transferable representations, we conduct a transfer experiment with the concept-guided image-based model. The model is trained on human face datasets (R & A) and evaluated on the cartoon face dataset IMAGEN, which contains 295 images labeled as angry, neutral, or happy (Schumann et al., 2010)(Lu et al., 2024). Without fine-tuning, the concept-guided model achieves 76.94% accuracy, surpassing the non-guided model (72.88%). These results demonstrate that the learned concepts extend beyond specific modalities and exhibit generalizability across domains.

Table 5: Comparison of unimodal model accuracy (%) under different concept-guided methods.

| Guidance | FED-RO | CMU-MOSI | | | CMU-MOSEI | | |
|---|---|---|---|---|---|---|---|
| | | Vision | Text | Audio | Vision | Text | Audio |
| Without guidance | 67.75 | 76.21 | 80.34 | 74.26 | 78.19 | 81.97 | 72.43 |
| Image-only | 73.75 | 83.24 | 83.47 | 79.86 | 82.96 | 83.17 | 77.98 |
| Text-only | 72.25 | 82.09 | 84.11 | 79.42 | 82.05 | 84.24 | 77.31 |
| Audio-only | 72.50 | 81.54 | 83.55 | 79.98 | 82.04 | 84.11 | 79.18 |
| Image + Text | 74.75 | 85.93 | 84.82 | 79.95 | 84.92 | 85.72 | 79.22 |
| Image + Audio | 74.75 | 85.34 | 84.55 | 80.12 | 84.65 | 85.03 | 79.98 |
| Text + Audio | 73.50 | 85.02 | 84.65 | 80.34 | 84.48 | 85.31 | 85.64 |
| Image + Text + Audio | **76.00** | **86.20** | **85.83** | **80.51** | **86.42** | **86.09** | **80.63** |
| Without selectivity | 70.18 | 80.42 | 82.17 | 78.04 | 80.32 | 82.31 | 76.25 |
| Without generalization | 72.00 | 83.74 | 84.28 | 78.56 | 85.34 | 84.88 | 79.22 |

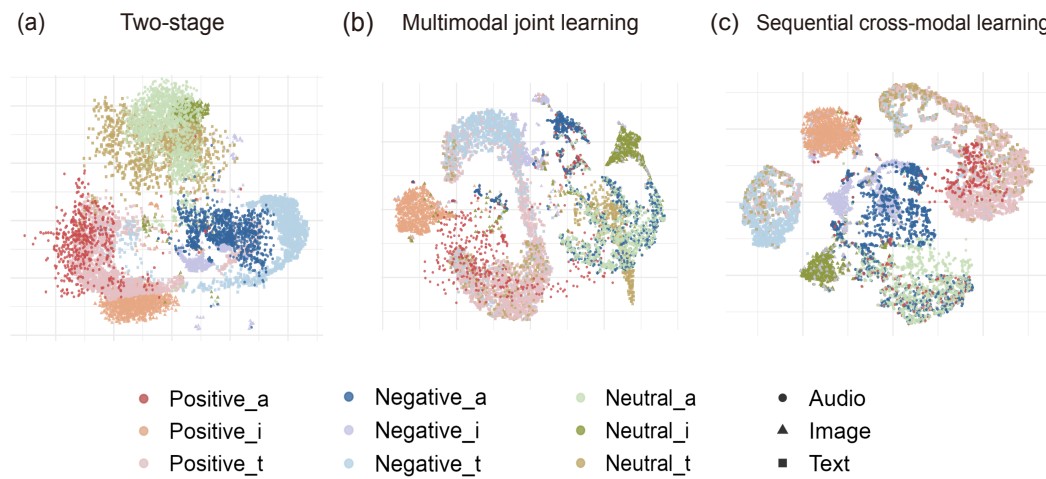

Figure 2: Exploring modality-independent emotion features extracted by the shared emotion encoder during supramodal concept learning. a: audio; i: image; t: text.

## 4.4 VISUALIZATION

**Emotion features vs. Non-emotion features** We adopt t-SNE(Van der Maaten & Hinton, 2008) to visualize the emotion and non-emotion features of each modality, illustrating the effectiveness of the disentanglement approach in supramodal concept learning. The shared emotion encoder clusters features by emotion labels with clear boundaries, while the modality-specific non-emotion encoders fail to distinguish emotions effectively (see Appendix F for details).

**Modality-independent emotion features are extracted by the shared emotion encoder.** We use t-SNE(Van der Maaten & Hinton, 2008) to visualize the emotion features of three modalities. The samples of images and audio are once again consolidated into three emotional categories consistent with those of the text. As shown in Fig. 2, after the two-stage training pipeline, the shared emotion encoder more effectively captures emotion features that are modality-independent, as features from the same emotion across different modalities cluster more cohesively, demonstrating the framework's ability to capture modality-independent emotion features.

## 5 CONCLUSION

In this work, we propose a brain-inspired framework to enhance emotion recognition by integrating supramodal emotion concepts as guiding principles. Our approach is motivated by neural mechanisms in the human brain, which form supramodal emotion concepts from multimodal experiences accumulated over development. Specifically, the framework constructs supramodal emotion concepts using a replay-based learning strategy and leverages them to regularize emotion recognition models. Experimental results validate the effectiveness of this approach, highlighting the crucial role of supramodal concept learning in guiding emotion recognition and demonstrating the potential of brain-inspired strategies to improve model robustness.

Despite these encouraging results, the current study primarily focuses on validating the framework through emotion classification. Future work should employ more comprehensive evaluation metrics to capture the nuanced effects of conceptual guidance. Extending the framework to dimensional emotion representations, such as valence-arousal scales, could provide a more granular understanding of how supramodal concepts influence emotion processing. Finally, evaluating the generalizability of supramodal concept learning to higher-order sociocognitive tasks—such as audiovisual speech integration or theory of mind—represents a promising direction for further advancing brain-inspired learning paradigms.

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

## A  DATASETS FOR EXPERIMENTS

### A.1  CURATED MULTIMODAL DATASET FOR SUPRAMODAL CONCEPT LEARNING

Despite the fact that our framework does not require paired image-text-audio datasets, multiple modalities are still essential for learning supramodal emotion concepts. However, our framework is more flexible in terms of data requirements, allowing the use of image, text, and audio datasets sourced independently from publicly available datasets.

For supramodal concept learning in this study, we use RAF-DB(Li et al., 2017b) as the image dataset, comprising approximately 30,000 facial images labeled with basic or compound expressions by 40 trained annotators. In our experiments, only images labeled with six basic expressions and neutral expressions are utilized. For the text modality, we adopt CMU-MOSI(Zadeh et al., 2016) and CMU-MOSEI(Zadeh et al., 2018b) datasets. CMU-MOSI consists of 93 opinion videos from YouTube vloggers, segmented into 2,199 opinion segments. Each segment is manually transcribed, and the start time of each sentence is annotated for alignment. Acoustic and visual features are extracted at sampling rates of 12.5 Hz and 15 Hz, respectively. CMU-MOSEI follows a similar format, consisting of 22,856 video segments of movie reviews sourced from YouTube, with acoustic and visual features sampled at 20 Hz and 15 Hz, respectively. Each text segment in both datasets is annotated as positive, neutral, or negative. For the audio modality, we employ three datasets: IEMOCAP(Busso et al., 2008), MELD(Poria et al., 2018), and RAVDESS(Livingstone & Russo, 2018). IEMOCAP includes 4,453 video segments annotated with categorical labels (six basic expressions and neutral) as well as dimensional labels (valence, arousal, dominance). MELD comprises 1,433 dialogues and 13,708 utterances from the TV series *Friends*. RAVDESS contains 7,356 speech and song recordings performed by 24 actors (12 male, 12 female) using a neutral North American accent. All three audio datasets provide annotations for the same seven emotion categories as the image datasets.

### A.2  FACIAL EMOTION DATASETS FOR CONCEPT-GUIDED UNIMODAL MODELS

In addition to RAF-DB, we also use AffectNet (Mollahosseini et al., 2017), the largest facial expression dataset to date that provides both categorical and valence-arousal annotations. AffectNet consists of an imbalanced training set and balanced validation and test sets. Additionally, we use FED-RO (Li et al., 2018), the first facial expression dataset featuring real-world occlusions. FED-RO was curated by collecting occluded images from Bing and Google search engines under appropriate licenses, followed by careful annotation by three independent annotators, resulting in 400 labeled images. All these datasets provide annotations for seven emotion categories.

For evaluating the image-based unimodal model, we train the model separately on the training sets of RAF-DB and AffectNet and assess its performance on their respective test sets. We also explore a combined training approach by merging the training sets of RAF-DB and AffectNet and evaluating the model on the independent test set of FED-RO.

### A.3  MULTIMODAL EMOTION DATASETS FOR CONCEPT-GUIDED UNIMODAL MODELS

To facilitate a fair comparison with previous multimodal fusion methods, we utilize the CMU-MOSI and CMU-MOSEI datasets, as these datasets provide paired vision-text-audio data derived from the same video segments — a common setup in prior work. In our framework, we can selectively use a single modality to train the concept-guided unimodal model while also conducting comparisons with algorithms that incorporate all three modalities. Since our focus in supramodal emotion concept learning is on emotion categories, and to align with prior methods under the same evaluation metrics, we follow the conventional practice of categorizing samples in both datasets as either positive or negative based on sentiment scores (greater than 0 or less than 0).

## B  DATA PREPROCESSING

In our study, we use RetinaFace(Deng et al., 2020) to detect facial regions and resize all emotional facial images to a uniform size of $224 \times 224 \times 3$, which are used both for supramodal concept learning and for downstream image-based models in the concept evaluation phase.

During supramodal concept learning, for the text data, sentences are first processed using the CLIP tokenizer, which applies byte-level Byte-Pair Encoding (BPE) with a maximum sequence length of 77 tokens. Sentences exceeding this limit are excluded, resulting in the removal of 224 sentences from CMU-MOSEI and 4 sentences from CMU-MOSI. For the audio data, since RAVDESS does not provide a predefined train–test split, samples from actor23 (male) and actor24 (female) are designated as the test set, while the remaining samples are used for training.

During supramodal concept evaluation, for the multimodal datasets used to train the concept-guided unimodal and multimodal models (*i.e.*, CMU-MOSI and CMU-MOSEI), we adopt the same pre-processing methods as previous works (Li et al., 2023). For the vision modality, each video frame is processed using Facet(Baltrušaitis et al., 2016) to extract the presence of 35 facial action units. For the text modality, we use a BERT-base-uncased pre-trained model(Devlin et al., 2019) to obtain a 768-dimensional hidden state as word features. For the acoustic modality, we apply COVAREP (Degottex et al., 2014) to extract 74-dimensional acoustic features. Tables 6-8 present the sample counts for each emotion category in each dataset used in this study.

Table 6: Sample counts for each emotion category in the facial emotion datasets used in this study.

| | Anger | Disgust | Fear | Happy | Neutral | Sad | Surprise | Total |
|---|---|---|---|---|---|---|---|---|
| RAF-DB | | | | | | | | |
| Train | 705 | 717 | 281 | 4772 | 2524 | 1982 | 1290 | 12271 |
| Test | 162 | 160 | 74 | 1185 | 680 | 478 | 329 | 3068 |
| AffectNet | | | | | | | | |
| Train | 24882 | 3803 | 6378 | 134415 | 74874 | 25459 | 14090 | 283901 |
| Test | 500 | 500 | 500 | 500 | 500 | 500 | 500 | 3500 |
| FED-RO | 53 | 51 | 58 | 59 | 50 | 66 | 63 | 400 |

Table 7: Sample counts for each emotion category in the text datasets used in this study.

| | Positive | Neutral | Negative | Total |
|---|---|---|---|---|
| CMU-MOSI | | | | |
| Train | 679 | 54 | 550 | 1283 |
| Test | 376 | 30 | 277 | 683 |
| CMU-MOEI | | | | |
| Train | 7966 | 3515 | 4678 | 16159 |
| Test | 2262 | 1022 | 1334 | 4618 |

## C  MORE IMPLEMENTATION DETAILS OF THE MULTIMODAL JOINT LEARNING

During this stage, we align $\boldsymbol{f}_I^{emo}$, $\boldsymbol{f}_T^{emo}$ and $\boldsymbol{f}_A^{emo}$ based on three general emotion categories: positive, neutral, and negative. However, since the original image and audio data include seven distinct emotion labels, we further examine whether the more granular negative emotions (fear, disgust, sadness, and anger) could still form modality-independent emotion clusters in $\boldsymbol{f}_I^{emo}$ and $\boldsymbol{f}_A^{emo}$ using t-SNE visualization.

After the two-stage training pipeline, images and audio still exhibit modality-independent characteristics across the seven emotion categories. As shown in Figure 3a, features are clustered by emotion, with no clear boundaries between audio and image samples within each category. In contrast, after single-stage training (Figures 3b and 3c), the shared emotion encoder retains modality-specific information. For most emotion categories including negative emotions, distinct boundaries are vis-

Table 8: Sample counts for each emotion category in the audio datasets used in this study.

| | **Anger** | **Disgust** | **Fear** | **Happy** | **Neutral** | **Sad** | **Surprise** | **Total** |
|---|---|---|---|---|---|---|---|---|
| IEMOCAP | | | | | | | | |
| Train | 776 | 2 | 33 | 530 | 1450 | 941 | 88 | 3820 |
| Test | 327 | 0 | 7 | 65 | 258 | 143 | 19 | 819 |
| MELD | | | | | | | | |
| Train | 1109 | 271 | 268 | 1743 | 4709 | 683 | 1205 | 988 |
| Test | 345 | 68 | 50 | 402 | 1256 | 208 | 281 | 2610 |
| RAVDESS | | | | | | | | |
| Train | 344 | 176 | 344 | 344 | 172 | 344 | 176 | 1900 |
| Test | 32 | 16 | 32 | 32 | 16 | 32 | 16 | 176 |

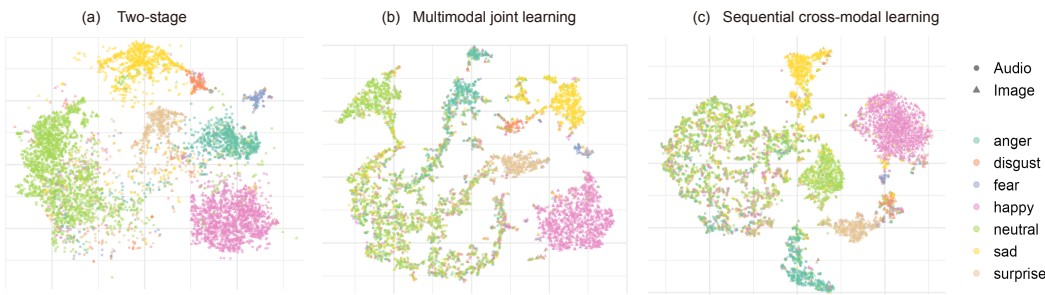

Figure 3: Assessing the extent to which the shared emotion encoder extracts modality-independent emotion features from images and audio across seven emotion categories during supramodal concept learning.

ible between image and audio features, suggesting that our method effectively preserves modality-independent emotion representations, even for fine-grained categories.

In the multimodal joint learning, we also explore aligning images and audio across seven emotion categories in addition to the three broad emotion classes across all modalities. Accordingly, we define two supervised contrastive loss functions based on Equation equation 3: $\mathcal{L}_{contrast}^{3-class}$ and $\mathcal{L}_{contrast}^{7-class}$. The former enforces coarse-grained alignment across three modalities, while the latter refines the alignment between images and audio at a finer, seven-class level. Thus, the overall loss function for the multimodal joint learning can be reformulated as follows:

$$\mathcal{L}_{joint} = \mathcal{L}_{task} + \beta_1(\mathcal{L}_{contrast}^{3-class} + \mathcal{L}_{contrast}^{7-class}) + \beta_2\mathcal{L}_{orth}. \tag{9}$$

The experimental results are shown in Table 9. When we add the seven-class alignment between images and audio, the performance shows no clear improvement compared to the three-class alignment. This may be due to the added alignment loss for the seven classes, which could shift the focus

Table 9: Comparison of the two-stage training pipeline and single-stage training on our curated multimodal emotion datasets.

| **Modality** | $\mathcal{L}_{contrast}^{3-class} + \mathcal{L}_{contrast}^{7-class}$ | | | $\mathcal{L}_{contrast}^{3-class}$ | | |
|---|---|---|---|---|---|---|
| | **sequential** | **joint** | **two-stage** | **sequential** | **joint** | **two-stage** |
| Image | 90.15 | 87.79 | **91.67** | 90.20 | 87.33 | 91.59 |
| Text | 72.31 | 70.05 | 74.96 | 73.67 | 70.25 | **75.42** |
| Audio | 56.12 | 53.21 | 59.95 | 56.84 | 53.81 | **60.97** |

away from effectively constraining emotion recognition. Nonetheless, these results also highlight the effectiveness of our framework, demonstrating that the shared emotion encoder learns modality-independent fine-grained emotion features without explicit seven-class alignment.

# D  THE PRETRAINING DATASETS USED FOR EVALUATING EACH METHOD ON FED-RO

Table 10: The pretraining datasets used for evaluating each method on FED-RO

| Method | Pre-train | FED-RO |
|---|---|---|
| VGG16Simonyan & Zisserman (2014) | ImageNet | 63.49 |
| ResNet18He et al. (2016) | ImageNet | 65.32 |
| gACNNLi et al. (2018) | R & A | 66.50 |
| SPWFA-SELi et al. (2020) | R & A | 67.25 |
| RANWang et al. (2020b) | MS-Celeb-1M | 67.98 |
| SCNWang et al. (2020a) | R & A | 68.24 |
| CRPNLu et al. (2024) | R & A | 71.00 |
| Ours | R & A | **76.00** |

# E  STATISTICAL VALIDATION OF CONCEPT GUIDANCE

We perform 10-fold cross-validation on the training sets of both MOSI and MOSEI. Results show that the concept-guided multimodal model significantly outperforms the unguided version on both datasets. These results confirm that the performance gains from our proposed method are statistically significant.

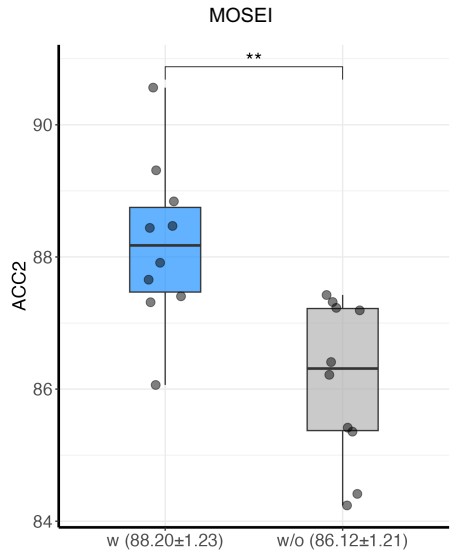
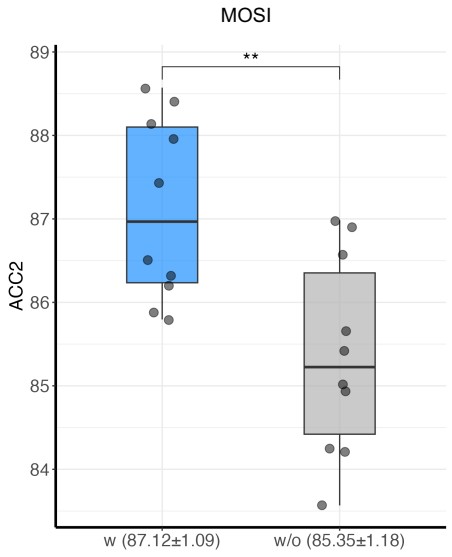

Figure 4: Performance comparison between concept-guided and unguided multimodal models on the MOSI and MOSEI datasets.

## F  EMOTION FEATURES VS NON-EMOTION FEATURES

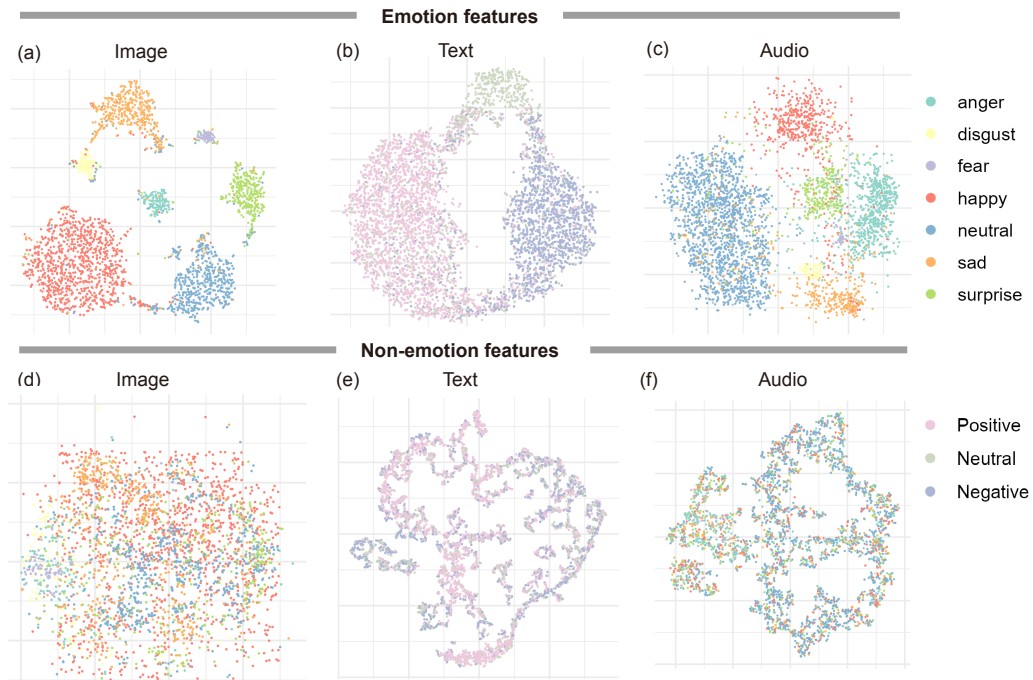

Figure 5: Visualization of the disentangled features for three modalities.

## G  DECLARATION OF LLM USAGE

Our work does not rely on LLMs for any part of the methodology or research process; they are used solely for polishing the writing.

