# OpenReview forum: "Human-like Supramodal Concept Learning Boosts Emotion Recognition"
_ICLR.cc/2026/Conference — ICLR 2026 Conference Withdrawn Submission_

### Official Review · Reviewer_p1Zm · 2025-10-14

**Soundness:** 2
**Presentation:** 2
**Contribution:** 2
**Rating:** 2
**Confidence:** 4

**Summary:**

This paper attempts to address challenges in multimodal emotion recognition by simulating the human brain's mechanisms of supramodal concept learning and hippocampal replay. The authors propose a two-stage training framework to learn modality-independent emotion representations, which are then used to construct so-called supramodal concepts to guide downstream tasks. While the experimental results are impressive, I have several major reservations regarding the paper's core motivation, the soundness of its methodology, and the rigor of its experimental design.

**Strengths:**

1. The method achieves competitive performance with a lightweight model, demonstrating excellent efficiency.

2. The methodological design relaxes the strict requirement for paired multimodal data, offering good practical flexibility.

**Weaknesses:**

1. The paper grounds its core motivation in human-like learning and hippocampal replay.  However, the implementation of these complex neuroscience concepts appears overly simplistic and superficial.
For instance, hippocampal replay is operationalized as retaining the top 20% correctly predicted samples per class and then averaging them to form a concept. This is, in essence, a method for constructing a selective prototype or centroid.
While potentially effective, forcibly linking this to the complex biological process of hippocampal replay seems to overstate the novelty of the mechanism.  A reviewer would expect a deeper, more mechanistic simulation rather than a mere terminological borrowing.

2. In the critical first stage of multimodal joint learning, to align image and audio (7 classes) with text (3 classes), the authors merge multiple distinct negative emotions (fear, disgust, sadness, and anger) into a single negative category.  This constitutes a severe loss of information. It means that the foundation for learning "modality-independent emotion representations" is built upon a coarse-grained label space that has lost significant nuance. This raises serious doubts about the quality and validity of the fine-grained "concepts" that are subsequently learned.

3. A fairer and more insightful comparison would be to benchmark their unimodal model against other state-of-the-art unimodal emotion recognition methods on the same datasets.

**Questions:**

1. Beyond a surface-level analogy, how does averaging high-confidence features substantively model the complex neural mechanism of hippocampal replay?

2. What is the theoretical motivation for the proposed two-stage training pipeline, which seems overly complex?

3. Why can't alignment and learning be achieved simultaneously via sequential learning to simplify the framework?

---

### Official Review · Reviewer_1wiN · 2025-10-25

**Soundness:** 1
**Presentation:** 2
**Contribution:** 1
**Rating:** 4
**Confidence:** 5

**Summary:**

This paper proposes a multimodal emotion recognition method that combines existing techniques, such as modality-specific and modality-shared representations, loss of orthogonality, alignment of equation 3, etc.

**Strengths:**

1. This paper proposes a learning strategy to construct supramodal emotion concepts across vision, text, and audio.
2. This paper uses the CLIP encoders to extract the multimodal features.

**Weaknesses:**

1. The novelty and contributions of this paper are too limited and marginal. In fact, this work is essentially almost a direct use of the existing methods. For example, modality-specific encoders, modality-shared encoders, and orthogonal loss are commonly used in existing works such as MISA[1], FDMER[2]. In particular, the alignment loss of Eq.3 is directly used of DMD[3].

[1] Misa: Modality-invariant and-specific representations for multimodal sentiment analysis.
[2] Disentangled representation learning for multimodal emotion recognition.
[3] Decoupled Multimodal Distilling for Emotion Recognition.

2. Comparative experiments are unfair. For example, the text feature of this work is from CLIP, but the text feature of other compared methods is from GloVe.Therefore, the experimental results have no credibility at all. The author should maintain the same experimental setup.
3. The paper heavily relies on neuroscientific terminology (vmPFC, hippocampal replay) to justify its approach, but the connection is tenuous and superficial. The model's "replay" mechanism is a gross oversimplification of a complex neural process. There is no evidence that sampling from high-confidence features mimics the selective, consolidative, and generative nature of hippocampal replay. The claim that the model's shared encoder mimics the vmPFC is an unsubstantiated analogy. The work does not provide any analysis (e.g., neural alignment) to support this claim beyond a simple reference to the brain region's function.

**Questions:**

Please see Weaknesses.

---

### Official Review · Reviewer_VUe3 · 2025-10-28

**Soundness:** 3
**Presentation:** 2
**Contribution:** 3
**Rating:** 4
**Confidence:** 4

**Summary:**

This paper proposes a human-like supramodal concept learning framework to enhance emotion recognition. Inspired by the human hippocampal replay mechanism and vmPFC supramodal encoding, it uses a decoupling framework of shared emotion encoder and modality-specific non-emotion encoder. Through two-stage training (multimodal joint and sequential cross-modal), it extracts modality-independent representations, constructs supramodal concepts to guide downstream models. Experiments verify its effectiveness on multiple datasets.

**Strengths:**

1. It closely integrates the human brain's supramodal emotion cognitive mechanism, simulates the selectivity and generalization of hippocampal replay to construct concepts, and adopts a Transformer architecture that conforms to the neural response of vmPFC, making the model design more in line with human emotional cognitive logic and enhancing the rationality and innovation of the method.

2. It covers multiple types of datasets (unimodal/multimodal, real/cartoon faces), conducts benchmark comparisons, ablation experiments and visualization verification, comprehensively verifying the framework's advantages in performance, robustness and generalization.

**Weaknesses:**

1. Although the paper proposes a brain-inspired supramodal concept learning framework, the technical novelty of the method itself is limited. The idea of extracting modality-independent emotional features is similar to that of MISA and FEDER, while the multimodal joint learning based on supervised contrastive loss resembles ConFEDE. Moreover, several existing emotion studies also address noisy labels by constructing soft-label constraints for consistency regularization, which is similar to this method.

2. Some references cited in the Introduction seem inaccurate or do not support the described claims. In particular, the references in the third paragraph of the Introduction fail to substantiate the discussion. Additionally, the statement “Transformer-based models outperform CNNs in capturing neural response patterns in mid-to-high-level brain regions, including the vmPFC (Caucheteux et al., 2023)” is questionable and should be carefully verified.

3. In the replay-inspired supramodal concept construction stage, the process of selecting the top 20% correctly predicted samples to build a high-confidence feature pool lacks sufficient explanation. It remains unclear how the confidence is measure, whether it is based on prediction logits, similarity scores, or another metric. And how the k high-confidence features are sampled from each modality pool.

4. Since the text modality uses only three sentiment labels (positive, negative, neutral), it is unclear whether the supramodal concept space is also three-dimensional. If so, the negative concept would merge emotions such as fear, disgust, sadness, and anger, which are semantically distinct. This coarse-grained abstraction may hinder fine-grained emotion recognition. It should clarify how supramodal concept evaluation benefits the final classification performance under this condition.

5. The baseline methods used for comparison are relatively outdated. Incorporating stronger and more recent multimodal or contrastive learning baselines would better demonstrate the competitiveness of the proposed approach.

6. The performance improvements reported are relatively modest, despite the use of large pretrained encoders (e.g., CLIP, CLAP) and multiple datasets for supramodal concept learning. It is thus unclear whether the gains stem from the proposed framework itself or from the advantages of pretrained models and data diversity.

**Questions:**

1. The current figure only illustrates the framework of supramodal concept learning. A figure that presents the entire pipeline, covering both the training and testing stage, would significantly improve clarity and reader comprehension.

2. Tables 3 and 4 lack explanations regarding the datasets used.

---

### Official Review · Reviewer_d6sL · 2025-11-01

**Soundness:** 3
**Presentation:** 3
**Contribution:** 4
**Rating:** 6
**Confidence:** 3

**Summary:**

The paper presents a brain-inspired framework for multimodal emotion recognition, introducing “supramodal emotion concepts” through a replay-based mechanism. The idea of decoupling emotion/non-emotion features and simulating hippocampal replay for concept construction is novel and intuitively appealing. The proposed model is technically sound, and extensive experiments demonstrate consistent performance gains over SOTA methods on multiple benchmarks.

**Strengths:**

+ The paper is well-constructed.
+ Clear motivation and well-designed two-stage training pipeline.
+ Lightweight yet high-performing architecture with LoRA fine-tuning.

**Weaknesses:**

- The replay idea works well in experiments, but the paper doesn’t clearly explain why it works. The method is mainly inspired by the hippocampal replay process in the brain, yet there’s no formal or mathematical framework behind it. For example, no optimization objective or information-theoretic view. Right now, the approach feels more like an intuitive, heuristic trick rather than something derived from solid principles. It would be much stronger if the authors could provide a clearer theoretical explanation for how replaying high-confidence samples improves cross-modal transfer.

- The phrase “supramodal emotion concept” is a key idea in the paper, but it’s never really defined in a precise way. It’s hard to tell how this concept differs from the usual “modality-invariant features” learned by contrastive or disentanglement-based models. Without a clear mathematical or algorithmic definition, the contribution feels a bit fuzzy, and it’s difficult to evaluate how general or novel the idea actually is.

**Questions:**

see weakness

---

### Note · Authors · 2025-11-19

I have read and agree with the venue's withdrawal policy on behalf of myself and my co-authors.